# Pemigatinib in Intrahepatic Cholangiocarcinoma: A Work in Progress

**Gennaro Gadaleta-Caldarola** [1], **Alessandro Rizzo** [2,*], **Vincenzo Dadduzio** [1], **Lucia Lombardi** [1], **Arianna Gadaleta-Caldarola** [3], **Stefania Infusino** [4], **Antonio Cusmai** [2], **Claudia Citrigno** [5] **and Gennaro Palmiotti** [2]

1 Unità Operativa Complessa di Oncologia Medica, Ospedale "Mons. A.R. Dimiccoli" Asl BT, Viale Ippocrate, 15, 70051 Barletta, Italy

2 Struttura Semplice Dipartimentale di Oncologia Medica per la Presa in Carico Globale del Paziente Oncologico "Don Tonino Bello", I.R.C.C.S. Istituto Tumori "Giovanni Paolo II", Viale Orazio Flacco 65, 70124 Bari, Italy

3 Facoltà di Medicina e Chirurgia, Università degli Studi di Siena, 53100 Siena, Italy

4 Unità Operativa Complessa di Oncologia Medica, Ospedale "SS. Annunziata", 87100 Cosenza, Italy

5 Facoltà di Medicina e Chirurgia, Università degli Studi Magna Graecia di Catanzaro, 88100 Catanzaro, Italy

* Correspondence: rizzo.alessandro179@gmail.com

**Abstract:** Cholangiocarcinoma (CCA) is the second most frequent primary liver cancer, following hepatocellular carcinoma (HCC). Progress in the molecular understanding of CCA has led to the development of several agents, including FGFR inhibitors, such as pemigatinib, whose approval has marked a new era in this hepatobiliary malignancy. However, a number of questions remain unanswered, including the development of secondary resistance and the role of combination therapies, including FGFR inhibitors. Herein, we specifically focus on the current challenges and future research directions of pemigatinib use in CCA patients.

**Keywords:** cholangiocarcinoma; biliary tract cancer; pemigatinib; FGFR2; intrahepatic cholangiocarcinoma; liver cancer

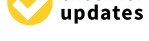

## 1. Introduction

Cholangiocarcinoma (CCA) represents the second most frequent primary liver cancer (PLC) following hepatocellular carcinoma (HCC), accounting for approximately 15% of all PLCs and 3% of gastrointestinal tumors worldwide [1,2]. Overall, the incidence and mortality rate of CCA continues to rise, mainly due to the presence of specific risk factors [3,4]. When surgically feasible, the standard of care treatment for CCA is surgical resection; however, most CCA patients present with unresectable or metastatic disease, and systemic chemotherapy represents the mainstay in this setting [5,6]. Until very recently, combination chemotherapy with cisplatin plus gemcitabine (CisGem) represented the frontline standard for advanced CCA, following the results of the practice-changing ABC-02 phase III trial that showed a median overall survival rate of approximately one year in CCA patients treated with the doublet [7,8]. However, the recently presented and published TOPAZ-1 trial has revealed a statistically significant improvement in overall survival in patients receiving the PD-L1 inhibitor durvalumab plus CisGem versus CisGem alone [9,10]. In addition, in recent years, we have witnessed an increasing interest in the targeting of molecular pathways involved in CCA tumorigenesis, and several targets of interest have been identified, including—among others—isocitrate dehydrogenase-1 (IDH-1), fibroblast growth factor receptor 2 (FGFR2), human epidermal growth factor receptor 2 (HER2), high microsatellite instability (MSI-H), and neurotrophic tropomyosin receptor kinase (NTRK) [11–17]. FGFRs represent a family of four receptor tyrosine kinases that play a crucial role in embryogenesis and organogenesis through several mechanisms, including cell migration, proliferation, and survival [18,19]. FGFR2 aberrations are estimated to be present in approximately 15–20%

of cases of intrahepatic cholangiocarcinoma (iCCA), which has led to the development and testing of drugs that target FGFR2 [20,21]. Among these, pemigatinib has reported practice-changing results in this setting and has been approved in several countries [22,23]. However, multiple questions remain unanswered, including the development of secondary resistance. Herein, we specifically focus on the current challenges and future research directions of pemigatinib use in CCA patients.

## 2. Fibroblast Growth Factor Receptors

FGFRs are a family of receptor tyrosine kinases, which are located on the cell membrane; once these receptors are activated by fibroblast growth factor (FGF), this link leads to the dimerization of FGFRs, with subsequent autophosphorylation of the intracellular kinase domain and activation of downstream pathways (Figure 1) [24,25].

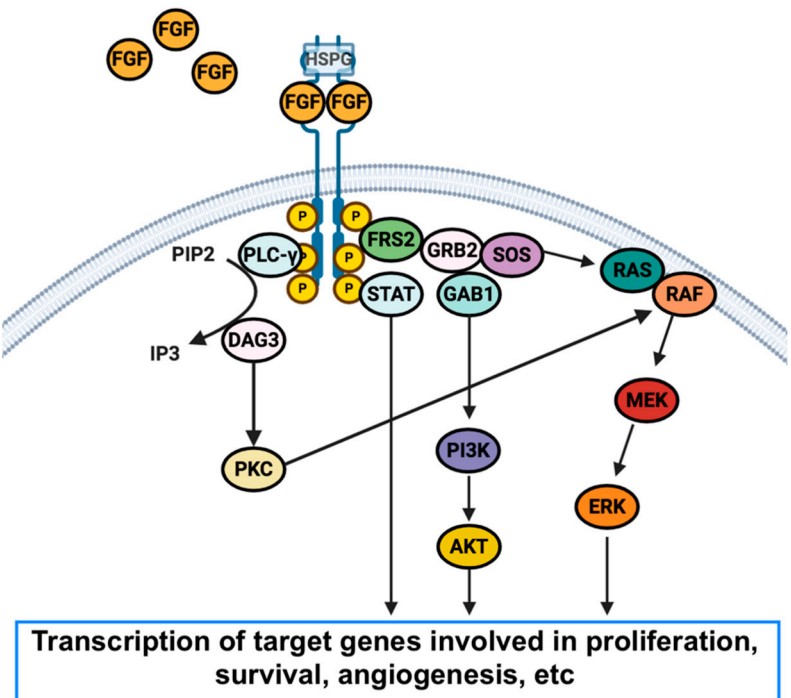

**Figure 1.** Schematic figure that represents the fibroblast growth factor receptor's (FGFR) structure, network, and alteration in cancer. Abbreviations: FRS2: fibroblast growth factor receptor substrate 2; HSPG: heparan sulfate proteoglycan; PLC-γ: phospholipase gamma; PIP2: phosphatidylinositol 4,5-bisphosphate; IP3: phosphatidylinositol 3,4,5-triphosphate; DAG: diacylglycerol; PKC: protein kinase C; GRB2: growth factor receptor-bound protein 2; GAB1: GRB2-associated-binding protein; MEK: MAPK/ERK kinase.

FGFRs differ in terms of anatomical distribution and function. For example, FGFR1 is mainly involved in cell cycle regulation and wound healing and FGFR2 and FGFR3 play also an important role in embryogenesis, while FGFR4 is involved in angiogenesis, glucose metabolism, and tissue repair [26,27]. Alterations including impaired FGFR signaling, gene amplification, FGFR fusion and mutations have been identified in several solid tumors, such as breast carcinoma, non-small cell lung cancer, gastric cancer, hepatocellular carcinoma, urothelial cancer, and melanoma [28–30]. In addition, FGFR fusions have been reported to play a crucial role in CCA, and these gene fusions have been identified in approximately 15% of cases of iCCA [31,32]. Typically, FGFR gene fusions are classified into two different types; type 1 fusions are caused by chromosomal translocations in hematological tumors, while type 2 fusions are related to chromosomal rearrangements in solid malignancies [33,34]. Fusion partners of FGFR2 in CCA include ROS, WAC, BICC1, AHCYL1, TACC3, and several others, and these partners are able to trigger auto-

dimerization in the absence of FGF and the activation of the downstream tyrosine kinase pathway [35,36]. Of note, FGFR2 gene fusions suggest a unique phenotype of iCCA, being associated with improved clinical outcomes, female gender, and younger age [37,38]. Moreover, these gene fusions are mutually exclusive with BRAF and KRAS mutations and have been reported almost exclusively in iCCA patients, while are very rare in other biliary tract cancer subgroups [39,40].

## 3. Pemigatinib

The FGFR1, FGFR2, and FGFR3 selective inhibitor pemigatinib is one of the two currently approved FGFR inhibitors for the treatment of previously treated metastatic CCA with FGFR2 gene fusion or translocation [41,42]. The role of pemigatinib in this setting was firstly explored in the phase I, two-part, FIGHT-101 trial [43]; part 1 of FIGHT-101 investigated the maximum tolerated dose and the pharmacological activity of pemigatinib according to serum phosphate elevation levels, while part 2 aimed to determine the recommended dose of pemigatinib in solid tumors with FGFR amplifications, translocations, or mutations. According to the results of this study, doses ranging from 1 to 20 mg showed dose-proportional increases in the maximum steady-state plasma drug concentration, reinforcing the importance of once-daily dosing [43]. In addition, no dose-limiting toxicity (DLT) was reported in part 1 of the trial, while the dose of 13.5 mg was recommended for part 2 of the trial. FIGHT-101 enrolled 128 cancer patients, including 16.4% patients with CCAs; partial response (PR) was observed in 9.4% of patients (n = 12), including five cases of CCAs. Hyperphosphatemia was the most common treatment-related emergent adverse event (75%), followed by stomatitis (29.7%), alopecia (28.1%), dysgeusia (25.8%), and dry mouth (25.8%) [43].

The subsequent phase II, open-label, multicenter FIGHT-202 trial has explored the role of pemigatinib (13.5 mg daily in three-week cycles) in three different cohorts of pretreated patients, including CCAs that harbored FGFR2 fusions or rearrangements (n = 107), those that harbored other FGF/FGFR aberrations (n = 20), and CCA patients without FGF / FGFR aberrations (n = 18) [44]. The primary endpoint of FIGHT-202 was ORR, which was 35.5% in the cohort of patients with FGFR2 fusions/rearrangements, with a median duration of response (DOR) of 7.5 months, median progression-free survival (PFS) of 6.9 months, and median overall survival (OS) of 21.1 months [44]. Conversely, median PFS and OS were 2.1 months and 6.7 months, respectively, in the cohort of patients that harbored other FGF / FGFR alterations, and 1.7 months and 4.0 months, respectively, in patients without FGF / FGFR aberrations [44]. Similar to FIGHT-101, hyperphosphatemia was the most frequent adverse event (55%), followed by alopecia (46%), dysgeusia (38%), diarrhea (34%), fatigue (31%), and dry mouth (29%). Grade 3 or higher adverse events were highlighted in 64% of the included patients, and grade 3–4 hypophosphatemia was the most common (12%) [44]. Similar results were also reported in the Chinese CIBI375A201 phase II trial, in which the ORR was 60% and median DOR was 8.3 months in previously treated advanced CCA. In a study that included 31 CCA patients with FGFR2 fusion or rearrangement, the median PFS was 9.1% [45].

Following the results of FIGHT-202, the United States Food and Drug Administration (FDA) granted accelerated approval for pemigatinib in previously treated CCA patients harboring FGFR2 fusions or rearrangements [46]. Of note, pemigatinib represented the first targeted therapy to be approved for the treatment of CCA and this marked a new era in this hepatobiliary malignancy. In the meantime, the ongoing, open-label, randomized FIGHT-302 phase III trial is currently comparing pemigatinib versus CisGem as first-line treatments in unresectable or metastatic CCA harboring FGFR2 rearrangement. PFS is the primary endpoint of the FIGHT-302 trial, with secondary endpoints including ORR, OS, DOR, safety, and disease control rate (DCR).

## 4. Open Questions and Future Research Avenues

In recent years, multiple phase II clinical trials that have explored FGFR inhibitors in CCA have reported interesting results, with some of these studies also leading to the FDA approval of pemigatinib and infigratinib in previously treated patients that harbor FGFR2 fusion or rearrangement [47,48]. Moreover, several phase III studies are currently ongoing, with the aim to compare novel FGFR inhibitors versus CisGem as first-line treatments, including the previously cited FIGHT-302, PROOF 301, and FOENIX-CCA3, by investigating pemigatinib, infigratinib, and futibatinib, respectively [49,50]. Regarding futibatinib, this agent has been tested in the single-arm, multicenter, phase II FOENIX-CCA2 trial that included previously treated CCA patients with FGFR2 gene fusions or other rearrangements. Futibatinib showed interesting and durable responses to futibatinib, with an ORR of 41.7% and median PFS of 8.9 months (Table 1) [50].

**Table 1.** A summary of some of the main results reported in clinical trials that investigated FGFR inhibitors in cholangiocarcinoma. Abbreviations: FGFR: fibroblast growth factor receptor; ORR: overall response rate; PFS: progression-free survival; TKI: tyrosine kinase inhibitor.

| Study | Phase | Agent | Mechanism of Action | ORR | Median PFS (Months) | Sample Size |
|---|---|---|---|---|---|---|
| FIGHT-202 | II | Pemigatinib | Selective FGFR1-3, reversible | 35.5% | 6.9 | 107 |
| NCT02150967 | II | Infigratinib | Selective FGFR1-3, reversible | 23.1% | 7.3 | 108 |
| FOENIX-CCA2 | II | Futibatinib | Selective FGFR1-4, irreversible | 41.7% | 8.9 | 103 |
| FIDES-01 | II | Derazantinib | TKI, reversible | 21.4% | 7.8 | 103 |

At the same time, the therapeutic scenario of CCA has witnessed some historical changes in parallel, as demonstrated by the recently published interim analysis of TOPAZ-1 that reported a statistically significant median OS improvement in CCA patients receiving CisGem plus durvalumab versus CisGem alone and established a new front-line standard in this disease. In patients that harbor druggable alterations, such as FGFR2 fusion or rearrangement, it is currently not clear whether FGFR inhibitors or CisGem plus durvalumab would be preferable. Moreover, some ongoing clinical trials are also investigating the eventual synergistic activity of immune-based combinations, including the FGFR inhibitor derazantinib plus the anti-PD-L1 agent atezolizumab (NCT05174650), in CCAs with FGFR2 fusion or rearrangement. Similarly, combination therapies with FGFR inhibitors plus anti-VEGFR agents are under assessment, due to the possible synergistic activity of these treatments and the role of FGFR in angiogenesis [51].

Another key point to consider is the development of resistance to FGFR inhibitors, as all the clinical trials that have explored these agents have shown a median PFS approximately ranging from 6 to 9 months. Several mechanisms have been associated with treatment resistance in patients who have received pemigatinib and other anti-FGFR agents, including feedback survival loop activation, gatekeeper mutations modifying drug access, or mutations impairing the binding site [52–55]. Of note, some reports have highlighted that covalent inhibitors may inhibit FGFR due to gatekeeper FGFR2 mutations, which impair the efficacy of pemigatinib and other FGFR inhibitors. Moreover, other combination treatments are being assessed, including those with anti-FGFR agents and PI3K/AKT/mTOR inhibitors (NCT04919642). In terms of treatment resistance, a key present and future challenge involves the use of circulating tumor DNA analysis in CCA patients, whose routine application may be of great importance for the detection and management of therapeutic resistance. In particular, the use of liquid biopsy may lead to the early identification of resistance mutations, as in the case of other cancer patients who receive targeted therapies—such as non-small cell lung cancer and colorectal cancer. Several questions remain unanswered, and further efforts are needed to develop methods, combinations, and agents that are able to overcome resistance to FGFR inhibitors.

**Author Contributions:** Conceptualization, G.G.-C. and A.R.; methodology, G.G.-C., V.D., L.L., A.G.-C., S.I., A.C., C.C., G.P.; software, G.G.-C. and A.R.; validation, all authors; formal analysis, G.G.-C. and A.R.; investigation, all authors; resources, G.G.-C. and A.R.; data curation, G.G.-C. and A.R.; writing—original draft preparation, G.G.-C. and A.R.; writing—review and editing, all authors; visualization, all authors; supervision, all authors; project administration, G.G.-C. and A.R.; funding acquisition, none. All authors have read and agreed to the published version of the manuscript.

**Funding:** This research received no external funding.

**Conflicts of Interest:** The authors declare no conflict of interest.

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
