# Peer review of "Pemigatinib in Intrahepatic Cholangiocarcinoma: A Work in Progress"

_curroncol, doi:10.3390/curroncol29100626_

Round 1

Reviewer 1 Report

Title: Pemigatinib in intrahepatic cholangiocarcinoma: A work in progress

This paper describes Pemigatinib in intrahepatic cholangiocarcinoma.

This paper including relatively large number of papers so the results may be highly reliable.

But there are some questions and the author is requested to add the descriptions according to comments as below.

Recently, U.S. FDA approves FGFR inhibitor, futibatinib for the treatment of previously treated, unresectable, locally advanced or metastatic intrahepatic cholangiocarcinoma.

The author should add details about futibatinib and FOENIX-CCA2 trial.

Author Response

Dear Reviewer,

Thank you for your comments and the time spent revising our paper. We added a paragraph regarding futibatinib and FOENIX-CCA2 trial, as suggested. Our changes have been reported in green colour.

We hope the revised paper will better suit the journal. 

Reviewer 2 Report

In this manuscript, Gadaleta-Caldarola et al. discuss the biology of FGFRs in various cancers. Then, the authors describe the FGFR inhibitor, Pemigatinib, and its effect on CCA patients' survival. Finally, the authors provide their questions and perspective regarding treatment options and resistance to FGFR inhibitors.

Overall, this perspective is well-written and provides a good summary of the use of FGFR inhibitors in intrahepatic cholangiocarcinoma. However, I have the following concerns and suggestions:

1- A figure (or table) summarizing the clinical trials of FGFR inhibitors will benefit the article's readers.

2- This article version seemed to me like a mini-review rather than a perspective. What do the authors propose to overcome FGFR inhibitor resistance? Or what experiments or research do they suggest to understand the molecular mechanism behind the resistance to FGFR inhibitors?  

Author Response

Dear Reviewer,

Thank you for the time spent revising our paper.

  1. We added a table (Table 1), as suggested.
  2. Thank you for this comment. In the revised manuscript, we included some additional paragraphs regarding present and future challenges in this setting, including ctDNA and liquid biopsy for CCA patients treated with targeted therapies. Our changes have been highlighted in red.

Thank you again. We hope the revised paper will better suit the journal.

Reviewer 3 Report

 The authors presented an interesting mini-review on the role of pemigatinib in cholangiocarcinoma. It is well written and complete but it is not a perspective study. There are at least 8 self-citations, I believe that the number may be decreased also in relation to the contents.

Author Response

Dear Reviewer,

Thank you for your suggestions.

We modified accordingly, in the revised paper.

Round 2

Reviewer 2 Report

In this version of the manuscript, Gadaleta-Caldarola et al., provided their perspective on how to detect and study FGFR inhibitors resistance in CCAs. They have also summarized the results of FGFR inhibitors clinical trials.  Overall the manuscript has improved and is suitable for publication as a perspective article.